# Legionella maintains host cell ubiquitin homeostasis by effectors with unique catalytic mechanisms

Jiaqi Fu [1,7], Siying Li[1,7], Hongxin Guan[2,7], Chuang Li[3,7], Yan-Bo Zhao[2,7], Tao-Tao Chen [2], Wei Xian [4], Zhengrui Zhang [5], Yao Liu [3], Qingtian Guan[1], Jingting Wang[2], Qiuhua Lu[2], Lina Kang[2], Si-Ru Zheng[2], Jinyu Li [6], Shoujing Cao[6], Chittaranjan Das [5], Xiaoyun Liu [4] ✉, Lei Song [1] ✉, Songying Ouyang [2] ✉ & Zhao-Qing Luo [3] ✉

The intracellular bacterial pathogen *Legionella pneumophila* modulates host cell functions by secreting multiple effectors with diverse biochemical activities. In particular, effectors of the SidE family interfere with host protein ubiquitination in a process that involves production of phosphoribosyl ubiquitin (PR-Ub). Here, we show that effector LnaB converts PR-Ub into ADP-ribosylated ubiquitin, which is further processed to ADP-ribose and functional ubiquitin by the (ADP-ribosyl)hydrolase MavL, thus maintaining ubiquitin homeostasis in infected cells. Upon being activated by actin, LnaB also undergoes self-AMPylation on tyrosine residues. The activity of LnaB requires a motif consisting of Ser, His and Glu (SHxxxE) present in a large family of toxins from diverse bacterial pathogens. Thus, our study sheds light on the mechanisms by which a pathogen maintains ubiquitin homeostasis and identifies a family of enzymes capable of protein AMPylation.

Successful pathogens use virulence factors to actively modulate host processes to evade cell-autonomous defense and immune detection. It is now recognized that the activity of virulence factors is a double-edged sword for the pathogen: on the one hand, these factors are essential for colonizing their hosts; on the other hand, cellular damage inflicted by pathogenic factors can be detected by specific receptors, leading to robust immune responses[1]. To minimize the chance of virulence factors being used as immune agonists, pathogens have evolved various regulatory mechanisms, including tight regulation of gene expression, precise targeting to specific cellular compartments, and/or the use of additional virulence factors to dampen the immune response[2–4].

In some scenarios, virulence factors may disrupt host cell homeostasis, thus making the cell less suitable for pathogen replication. One such example is the intracellular bacterial pathogen *Legionella pneumophila*. This bacterium creates an intracellular niche permissive for its replication utilizing a large cohort of effectors to modulate host processes by diverse biochemical activities[5]. Among the scores of

[1]Department of Respiratory Medicine, Center for Infectious Diseases and Pathogen Biology, Key Laboratory of Organ Regeneration and Transplantation of the Ministry of Education, State Key Laboratory for Diagnosis and Treatment of Severe Zoonotic Infectious Diseases, The First Hospital of Jilin University, Changchun, China. [2]Key Laboratory of Microbial Pathogenesis and Interventions of Fujian Province University, the Key Laboratory of Innate Immune Biology of Fujian Province, Biomedical Research Center of South China, Key Laboratory of OptoElectronic Science and Technology for Medicine of the Ministry of Education, College of Life Sciences, Fujian Normal University, Fuzhou, China. [3]Department of Biological Sciences, Purdue University, West Lafayette, IN, USA. [4]Department of Microbiology, NHC Key Laboratory of Medical Immunology, School of Basic Medical Sciences, Peking University Health Science Center, Beijing, China. [5]Department of Chemistry, Purdue University, West Lafayette, IN, USA. [6]College of Chemistry, Fuzhou University, Fuzhou, Fujian, China. [7]These authors contributed equally: Jiaqi Fu, Siying Li, Hongxin Guan, Chuang Li, Yan-Bo Zhao. ✉e-mail: xiaoyun.liu@pku.edu.cn; l.song@139.com; ouyangsy@fjnu.edu.cn; luoz@purdue.edu

effectors that target the ubiquitin network, members of the SidE effector family pose unique challenges to host cells. These effectors catalyze protein ubiquitination by a mechanism involving the production of ADP-ribosylated ubiquitin (ADPR-Ub) as the reaction intermediate using their monoADP-ribosyltransferase (mART) activity[6]. ADPR-Ub is then used to modify substrate proteins by their phosphodiesterase (PDE) domains[7,8]. These two reactions are uncoupled, and ADPR-Ub may accidentally release into the cytosol of infected cells. Furthermore, the reversal of SidEs-induced ubiquitination by DupA and DupB produces phosphoribosyl ubiquitin (PR-Ub)[9,10]. Neither ADPR-Ub nor PR-Ub can be used in canonical ubiquitination reactions[7], and their accumulation may interfere with key signaling cascades in infected cells and ultimately impede bacterial replication. These challenges necessitate mechanisms to maintain ubiquitin homeostasis in infected cells.

Here we show that the bacterium utilizes two sequential reactions catalyzed by additional effectors to convert PR-Ub into functional ubiquitin. After being activated by the host protein Actin, LnaB reduces PR-Ub to ADPR-Ub, which is hydrolyzed into ADP-ribose and native ubiquitin. We also show that LnaB represents a large family of bacterial toxins potentially involved in cell signaling by protein AMPylation by a mechanism that requires catalysis induced by a motif consisting of Serine, Histidine, and Glutamate.

## Results

### The macro domain protein MavL is an (ADP-ribosyl)hydrolase against ADPR-Ub

We attempted to identify additional *L. pneumophila* effectors involved in ubiquitin signaling using biotin-mediated proximal labeling (Turbo ID)[11], the BirA*-Ub fusion (BirA*, the promiscuous BirA mutant)[11] was expressed in *L. pneumophila* and biotinylated proteins were identified by mass spectrometry (Supplementary Fig.1a). Not surprisingly, a number of Dot/Icm effectors known to be involved in ubiquitin signaling were identified. In addition, a few effectors of unknown function were detected with high confidence (Supplementary Fig. 1b). Among these, MavL(Lpg2526) has been studied in a recent study which reveals that this protein binds ADP-ribose and structural overlay with poly-(ADP-ribose) glycohydrolases (PARGs) identified two aspartate residues potentially involved in catalysis[12]. This study also found that MavL interacts with the mammalian ubiquitin-conjugating enzyme UBE2Q1, suggesting a role in ubiquitin signaling[12]. In light of these observations and the fact that MavL was identified in our screening strategy based on ubiquitin interaction, we further pursued the hypothesis that MavL acts on proteins that have been modified by ubiquitination, ADP-ribosylation, or both. That APDR-Ub is inactive in canonical ubiquitination reactions (Supplementary Fig. 1c) prompted us to examine whether it is a substrate of MavL. Indeed, recombinant MavL effectively hydrolyzed ADPR-Ub into ADPR and ubiquitin, which is in contrast to DupA and DupB (Fig. 1a). Like most Dot/Icm effectors, MavL is not required for bacterial intracellular growth (Supplementary Fig. 1d).

To understand the catalytic mechanism of the ADP-ribosylhydrolase (ARH) activity of MavL, we screened its active truncation mutants and succeeded in solving the structure of the MavL$_{40-404}$-ADPR complex at a resolution of 2.38 Å, which revealed that several pairs of hydrogen bonds and a π-π stacking interaction provided by surrounding amino acids or water molecules are involved in interactions between MavL and ADPR (Supplementary Fig. 1e). The negatively charged residues D315, D323, and D333 likely form a catalytic loop (β8-α8) that directly interacts with the hydroxyl groups of ribose by hydrogen bonding (Supplementary Fig. 2a). Mutational analysis indicated that all three residues are critical for the ARH activity of MavL against ADPR-Ub (Fig. 1b). The role of D323 and D333 in catalysis has been predicted in an earlier structural study of MavL[12]. Furthermore, the mutants MavL$_{D315A}$ and MavL$_{D323A}$ displayed higher affinity toward ADPR-Ub and ADPR (Fig. 1c and Supplementary Figs. 2b, 3, 4). Further

efforts using these mutants allowed us to obtain the structure of the MavL$_{40-404(D315A)}$-ADPR-Ub complex at a 1.93 Å resolution (Fig. 1d, **left**). The combined surface area of the interface between MavL and ubiquitin is approximately 867.6 Å$^2$. In the complex, ADPR-Ub engages MavL in such a way that the ADPR moiety linked to the side chain of R42 of ubiquitin is inserted into the active-site pocket and the binding is coordinated by several pairs of hydrogen bonds provided by G223, G225, C226, F227, K236, P264, N322, D323, T331, D332, and D333 of MavL with D39 and R42 of ubiquitin (Supplementary Fig. 2c). E107 and D323 of MavL hold R42 of ubiquitin in a position suitable for catalysis via hydrogen bond interactions (Fig. 1d, **middle**). In addition, the side chain of D323 points to the 1"-OH of the ADPR moiety and the -NH2 of R42 in ubiquitin at a distance of 2.65 Å and 2.69 Å, respectively, thus is most likely the residue key for catalysis (Fig. 1d, **middle**). Unexpectedly, the N-glycosidic bond between ADPR and R42 of ubiquitin was cleaved in our structure, which may be caused by the residual activity of the D315A mutant. (Fig. 1d, **right**).

Comparison of the structures of apo-MavL, MavL-ADPR, and MavL$_{40-404(D315A)}$-ADPR-Ub revealed that the side chains of several residues, including F105, C226, Y232, and Q330 are involved in the formation of the activity pocket of apo-MavL that faces outwards, keeping the binding pocket in an open state (Fig. 1e). Among these, the side chains of C226 and Q330 form steric hindrance with the side chains of D39 and Q40 on ubiquitin loop1 and E51 and D52 on ubiquitin loop2, respectively, preventing ubiquitin from binding to apo-MavL(Supplementary Fig. 5a). In addition, we observed four water molecules around the ADPR moiety, two of which (H$_2$O-1 and H$_2$O-2) are also found in other macro domain ARHs (Supplementary Fig. 5b). These two water molecules form a hydrogen bond network with the α-phosphate of ADPR and the O1" site of the distal ribose group, wherein the α-phosphate activates H$_2$O-1 to initiate a nucleophilic attack on the O1" glycosidic bond, leading to its cleavage (Fig. 1d, **middle**).

We examined substrate specificity of MavL using several ADP-ribosylated proteins, including ADPR-Ub$_{T66}$ produced by CteC of *Chromobacterium violaceum*[13], ADPR-Actin induced by SpvB of *Salmonella enterica*[14], ADPR-ANT1 catalyzed by Ceg3 of *L. pneumophila*[15], and ADPR-PARP1 catalyzed by Sirt6[16]. In addition to ADPR-Ub produced by SidEs, ADP-ribosyl hydrolysis against ADPR-ANT1 and ADPR-Actin also detectably occurred (Supplementary Fig. 2d). We further examined the physiological role of MavL by probing ADPR-Ub in cells infected with relevant *L. pneumophila* strains. ADPR-Ub was detected in cells infected by wild-type *L. pneumophila* and its level was elevated in samples infected with the Δ*mavL* mutant, and complementation with MavL but not the inactive mutant MavL$_{D323A}$ restored the phenotype (Fig. 1f), indicating that MavL functions to reduce cellular ADPR-Ub. Our attempt to determine the distribution of MavL in cells infected with *L. pneumophila* by immunostaining was not successful, probably due to low protein abundance or the quality of our antibodies. Importantly, infection of cells transfected to express HA-MavL revealed a clear accumulation of protein on the Legionella-containing vacuole (LCV) in a manner that required a functional Dot/Icm system (Supplementary Fig. 6), suggesting that the effector mainly acts on the cytoplasmic surface of the bacterial phagosome.

### The effector LnaB is an adenylyltransferase that converts PR-Ub into ADPR-Ub

Efficient conversion of ADPR-Ub into ADPR and ubiquitin by MavL suggests that this enzyme functions to return modified ubiquitin to its native form. Yet, MavL cannot remove the phosphoribosyl group from PR-Ub (Supplementary Fig. 2e), we thus considered the possibility that PR-Ub is first converted into ADPR-Ub prior to hydrolysis by MavL.

Dot/Icm effectors of relevant functions often are encoded by genes of close proximity on the chromosome[5,17]. The gene upstream of *mavL* is *mavK*(lpg2525), which appears to harbor an F-box motif known to be involved in ubiquitination[18]. The genes coding for LnaB (Lpg2527)

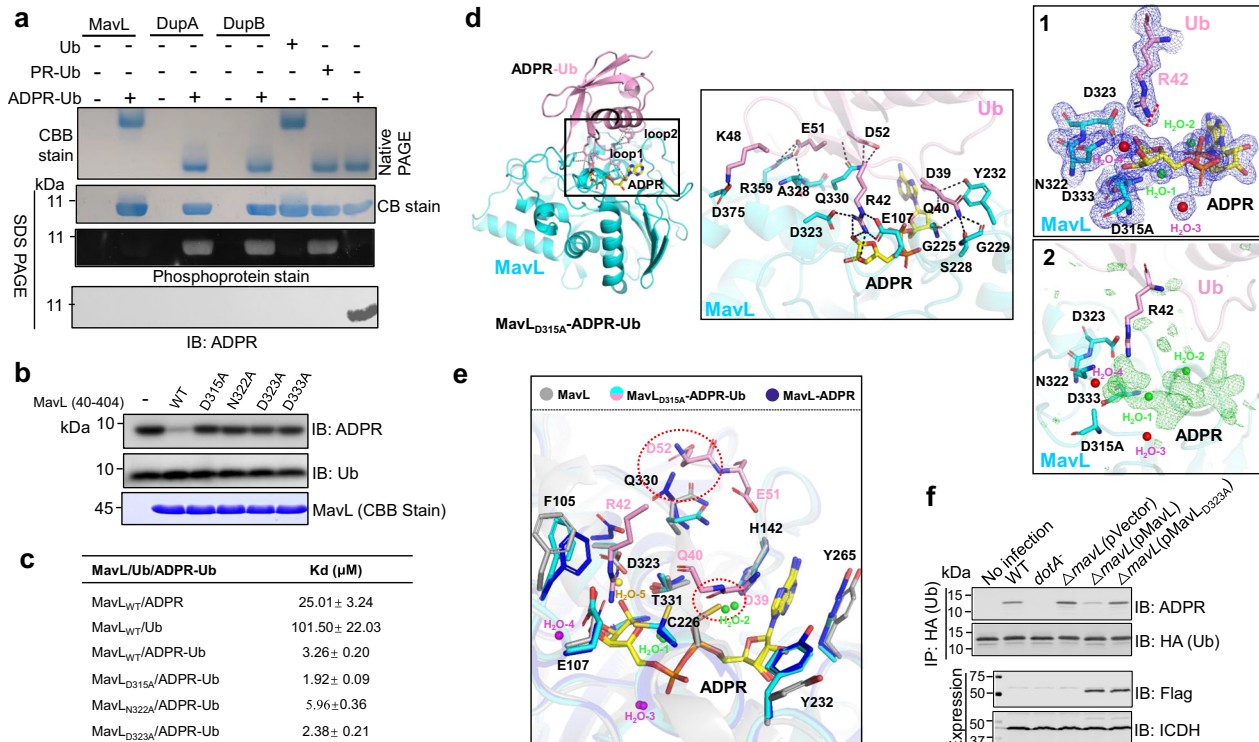

**Fig. 1 | MavL is a macro domain protein that converts ADPR-Ub into ADP-ribose and ubiquitin. a** Hydrolysis of ADPR-Ub into ADPR and Ub by MavL DupA or DupB. Recombinant proteins were incubated with ADPR-Ub, and the production of native Ub was detected by native polyacrylamide gel electrophoresis (upper panel). Native Ub, PR-Ub, and ADPR-Ub were loaded separately as controls. Identical samples separated by SDS-PAGE were detected by CBB staining, phosphoprotein stain, or immunoblotting with an ADPR-specific antibody (lower three panels). **b** Mutational analysis of residues important for the de-ADP ribosylation activity of MavL. Recombinant MavL or its mutants were incubated with ADPR-Ub, and the reduction of the reactant was detected by immunoblotting with an ADPR-specific antibody. **c** Binding of Ub, ADPR-Ub, and ADPR to MavL$_{40-404}$ or its mutants. Binding affinity was evaluated using a low-volume Nano ITC set at 20 °C. **d** Ribbon diagram representation of the MavL$_{(40-404)D315A}$-ADPR-Ub complex. Ub, ADPR, and MavL are colored in pink, yellow, and cyan, respectively. The recognition of ADPR-Ub by MavL, as well as the interactions between the two proteins, are shown in the middle panel. The 2Fo−Fc (blue) and Fo−Fc (green) electron-density maps of the key

residues of MavL, Ub, and water surrounding the ribose moiety involved in forming the catalytic center are contoured at the 1.0σ and 3.0σ levels and shown in 1 and 2 of the right panel, respectively. It represents the transient state of the substrate catalyzed by MavL. The N-glycosidic bond between the side chain of R42 in Ub and the ADPR moiety was cleaved, as indicated by the red dashed lines and arrow. **e** The overall structure of MavL$_{(40-404)D315A}$-ADPR-Ub and its comparison to Apo MavL (gray) and MavL-ADPR (blue). The conformational changes that contribute to the opening of the catalytic pocket to facilitate the binding of ADPR-Ub and the subsequent reaction were shown. Residues that cause steric hindrance between Apo MavL and Ub were marked with red dashed circles. **f** MavL reduces the level of ADPR-Ub in infected cells. The indicated *L. pneumophila* strains were used to infect cells expressing 3xHA-Ub, and the accumulation of ADPR-Ub was detected after HA antibody immunoprecipitation. Expression of Flag-MavL and its mutant was detected with Flag antibody and isocitrate dehydrogenase (ICDH) was probed as a loading control.

and the deubiquitinase Lem27 (Lpg2529) (also known as LotC)[19] are separated by a gene predicted to code for an α-amylase (Supplementary Fig. 7a). We then examined the hypothetical proteins, MavK and LnaB, for the ability to convert PR-Ub into ADPR-Ub. Incubation of recombinant MavK[18] or LnaB with PR-Ub and ATP did not detectably produce ADPR-Ub (Supplementary Fig. 7b). Some *L. pneumophila* effectors are known to require host co-factors for their activity[20-23], we thus added lysates of mammalian cells to these reactions and found that native, but not heat-treated lysates enabled LnaB to produce ADPR-Ub from PR-Ub (Fig. 2a). Thus, LnaB has the capacity to convert PR-Ub into ADPR-Ub in the presence of a eukaryotic cell-specific molecule, which likely is a protein due to its sensitivity to heat treatment.

To identify the host factor required for LnaB activity, we identified interacting proteins by immunoprecipitation from lysates of cells transfected to express Flag-LnaB by mass spectrometric analysis. Among the proteins specifically enriched by Flag-LnaB, Actin was identified as the most differed and abundant hit (Fig. 2b, Supplementary Fig. 7c and Supplementary Dataset 1). In line with these results, LnaB and Actin formed a complex in cells that was readily detectable by immunoprecipitation and by analytic ultracentrifugation

(Supplementary Fig. 7d, e). ITC analysis revealed that these two proteins bind each other with a K$_d$ of ~1.24 μM (Fig. 2c). More importantly, inclusion of Actin in reactions containing PR-Ub, ATP and LnaB led to the production of ADPR-Ub (Fig. 2d). PR-Ub differs from ADPR-Ub only by an adenosine monophosphate (AMP) moiety (Fig. 2e), suggesting that LnaB catalyzes a reaction at the α phosphate center of ATP to transfer the AMP moiety onto PR-Ub. Indeed, the inclusion of $^{32}$P-α-ATP in the reaction led to the production of radio-labeled ADPR-Ub, again in an Actin-dependent manner (Fig. 2f).

ATP analogs containing a cleavable α phosphate, including adenylyl-imidodiphosphate (AMPPNP) and adenosine 5′-(γ-thio)triphosphate (ATPγS), supported the full activity of LnaB. In line with its partially susceptible α phosphate[24], ATPαS also supported the activity (Fig. 2g). In contrast, adenosine 5′-(α, β-methylene)triphosphate (ApCpp), which has an uncleavable α-site, was unable to serve as the nucleotide donor (Fig. 2g).

Similar results were obtained when the product of the reaction was analyzed by mass spectrometry, which revealed that the tryptic peptide of ubiquitin (-E$_{34}$GIPPDQQRLIFAGK$_{48}$-) derived from PR-Ub in which R42 was modified by phosphoribosylation had been converted into ADP-ribosylation after incubation with ATP, LnaB, and Actin

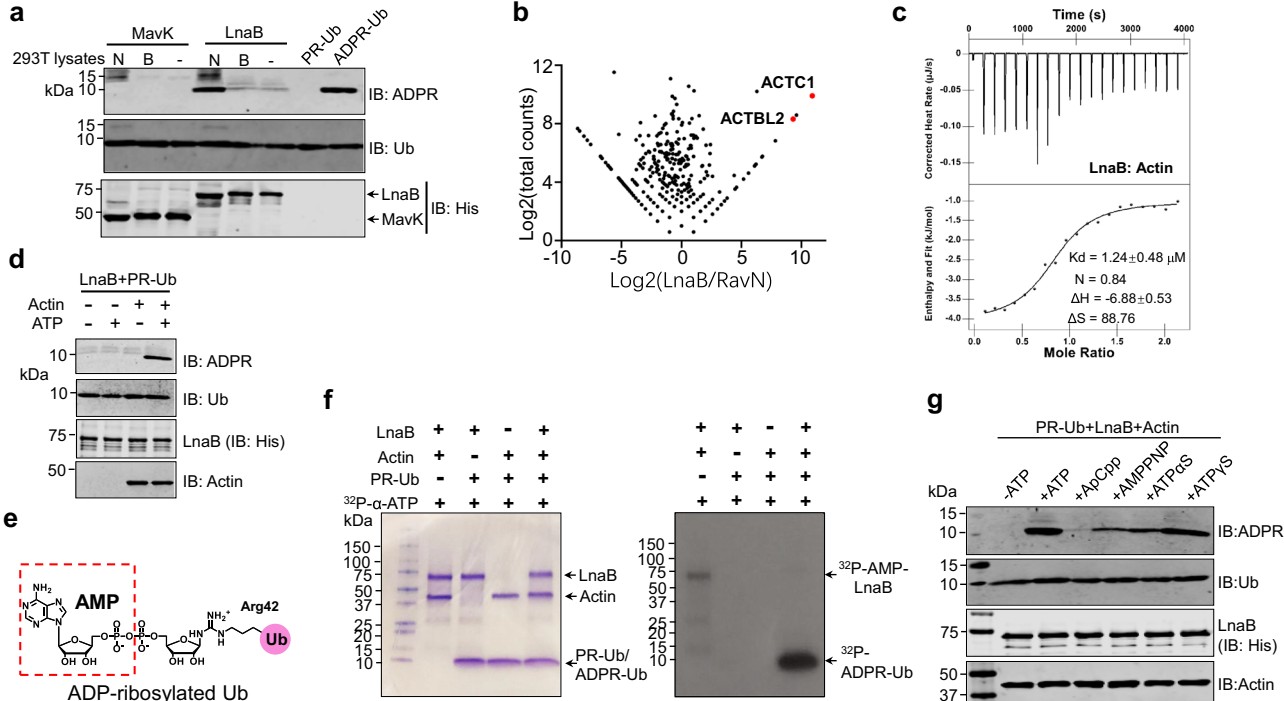

**Fig. 2 | Conversion of PR-Ub into ADPR-Ub by LnaB requires Actin as a co-factor. a** Native lysates of mammalian cells activate LnaB. Native (N) or boiled (B) lysates of 293HEK cells were added to reactions containing PR-Ub, MavK, or LnaB, and the production of ADPR-Ub was detected by immunoblotting. **b** Identification of Actin as a LnaB-binding protein. Flag-LnaB expressed in HEK293T cells was isolated by immunoprecipitation and the bound proteins were identified by mass spectrometry. Similarly obtained samples with Flag-RavN were used as a control. alpha Actin (ACTC1) and beta Actin-like 2 (ACTBL2) were among the most abundant proteins identified. **c** Interactions between LnaB and Actin measured by ITC. Raw ITC curves (top panel) and binding isotherms with fitting curves (bottom panel) of LnaB titration by Actin. The thermogram is a monophasic curve with an inflection point at a molar ratio of 0.84. The binding affinity is approximately 1.24 µM, and the stoichiometry is 1:1 of Actin:LnaB. The thermodynamic parameters were also shown, $\Delta H$: − 6.88 kJ•mol$^{-1}$ and $\Delta S$: 88.75 J•mol$^{-1}$•K$^{-1}$. **d** LnaB and actin utilize ATP to convert PR-Ub into ADPR-Ub. Actin was added to a subset of reactions containing

LnaB and PR-Ub. Samples separated by SDS-PAGE were probed for ADPR-Ub (upper panel), ubiquitin, LnaB, or Actin by immunoblotting with antibodies specific to each protein or its epitope tag. **e, f.** LnaB transfers the AMP moiety of ATP to PR-Ub. The chemical structure of ADPR-Ub with the AMP moiety added to the phosphate group on PR-Ub being highlighted (dashed box) (**e**). $^{32}$P-α-ATP was added to the indicated reactions and incubated at 37 °C for 1 h. Samples separated by SDS-PAGE were detected by CBB staining (left) and autoradiograph, respectively. Note the presence of self-modified LnaB in the reaction without PR-Ub (**f**). **g** ATP analogs with a cleavable α phosphate support LnaB activity. Samples of reactions receiving the indicated ATP analogs were resolved by SDS-PAGE and ADPR-Ub and the reactants were detected by immunoblotting by antibodies specific for ADPR, Ub, LnaB, or Actin. Note that ApCpp is uncleavable at the α position and thus did not support the activity of LnaB. In each case, similar results were obtained in at least three independent experiments.

(Fig. 3a–c). Importantly, ADPR-Ub produced by LnaB from PR-Ub can be used in ubiquitination induced by the PDE activity of SidEs. Incubation of the product with SdeA$_{E/A}$ (a SdeA mutant without mART activity but retaining its PDE function) led to Rab33b ubiquitination at levels comparable to reactions receiving native ADPR-Ub (Fig. 3d). Furthermore, ubiquitin produced from PR-Ub by LnaB and MavL was active in canonical ubiquitination reactions (Fig. 3e).

## LnaB utilizes an S-HxxxE motif to catalyze the conversion of PR-Ub into ADPR-Ub

We explored the mechanism of action of LnaB using PSI-BLAST[25] searches to identify proteins that may have common functional motifs, which revealed that LnaB belongs to a family of toxins of at least 103 members that harbor a conserved S-HxxxE (x, any amino acid) motif of unknown biochemical activity (Supplementary Fig. 8). Notably, the lengths of the space between the conserved Serine and Histidine residues vary greatly among members of the protein family (Supplementary Fig. 8a). These proteins are encoded by diverse bacterial pathogens of a wide range of hosts, particularly a large set of proteins categorized as Making caterpillar floppy (MCF) toxins found in insect pathogens (Supplementary Fig. 8b)[26]. Sequence alignment revealed that in LnaB, the S-HxxxE motif is composed of S261, H305, and E309 (Supplementary Fig. 8a). We examined the role of this predicted motif

in the activity of LnaB by creating substitution mutants for each of these residues and testing their activity in converting PR-Ub into ADPR-Ub. Mutations in S261, H305, or E309 completely abolished the enzymatic activity (Fig. 4a, b).

To determine the role of LnaB during *L. pneumophila* infection, we employed mass spectrometric analysis to probe PR-Ub in cells infected with relevant bacterial strains. Very weak signals of PR-Ub were detected in cells infected with wild-type bacteria, but it became abundant in cells infected with the Δ*lnaB* mutant. Complementation with LnaB but not the LnaB$_{S261A}$ mutant rendered PR-Ub undetectable (Fig. 4c), indicating that LnaB functions to eliminate PR-Ub in infected cells.

We also examined the distribution of LnaB in cells infected with *L. pneumophila* but were not able to detect signals using our antibodies specific to this protein. Yet, results from infection of cells transfected to express 4Flag-LnaB indicated that the protein was enriched on the LCV and such enrichment was dependent upon an active Dot/Icm system as vacuoles containing the *dotA* mutant did not detectably recruit Flag-LnaB (Supplementary Fig. 9a). These results suggest that similar to MavL, LnaB likely acts on the surface of the bacterial phagosome.

Similar to earlier experiments, in cells infected with wild-type *L. pneumophila*, ADPR-Ub is detected, and adding LnaB led to a slight

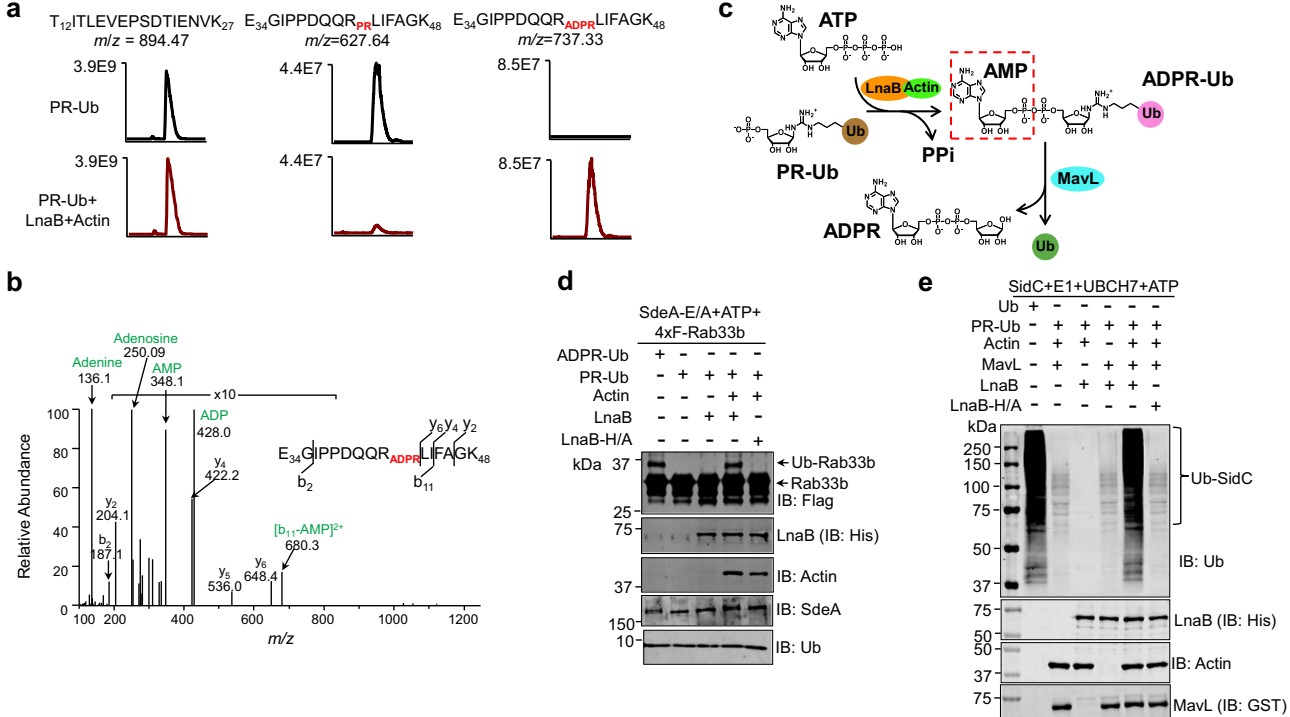

**Fig. 3 | LnaB and MavL sequentially convert PR-Ub into ADPR and active ubiquitin. a, b** Detection of LnaB-mediated conversion of PR-Ub into ADPR-Ub by mass spectrometric analysis. Excised protein bands from SDS-PAGE gels corresponding to PR-Ub prior to the reaction or ADPR-Ub after incubation with ATP, LnaB, and Actin were digested with trypsin and analyzed by mass spectrometry. A reference fragment $T_{12}$ITLEVEPSDTIENVK$_{27}$ was present in both samples with similar abundance (a left panel). The abundance of the fragment with PR-modified R42 was high in the PR-Ub samples but became almost undetectable after reaction with LnaB, ATP, and Actin, which was accompanied by the increase of the ADPR-modified fragment. An MS/MS spectrum indicating ADPR modification of R42 was shown in (**b**). **c** A reaction scheme depicting the conversion of PR-Ub into ubiquitin by LnaB and MavL. The AMPylation activity of LnaB first converts PR-Ub into ADPR-Ub, which is further reduced into ADP-ribose and ubiquitin by MavL. The AMP moiety defined by a dashed line rectangle indicates the chemical group added to PR-Ub by LnaB. **d** The use of ADPR-Ub produced from PR-Ub by LnaB in protein modification by the phosphodiesterase (PDE) activity of SdeA. PR-Ub was incubated in the indicated reactions and the ability to ubiquitinate Rab33b was detected by the formation of higher MW species detected by immunoblotting with the Flag-specific antibody. Native ADPR-Ub was included as a control (1st lane). **e** Conventional ubiquitination by ubiquitin produced by MavL and LnaB from PR-Ub. A series of reactions containing PR-Ub and combinations of relevant proteins were allowed to proceed for 1 h at 37 °C. The products were boiled for 5 min at 95 °C, and a cocktail containing E1, E2, SidC (E3), and ATP was added, self-ubiquitination of SidC was detected by immunoblotting with a ubiquitin-specific antibody.

increase in its abundance (Fig. 4d). Importantly, ADPR-Ub was not detectable in cells infected with the Δ*lnaB* mutant but became abundant after adding recombinant LnaB (Fig. 4d), which is consistent with the notion that PR-Ub was accumulated in these cells. PR-Ub accumulation caused by infections with strain Δ*lnaB* can be reversed by complementation with LnaB but not the LnaB$_{S261A}$ mutant (Fig. 4d). LnaB-dependent elimination of PR-Ub in infected cells by converting it into ADPR-Ub with recombinant LnaB was also determined. We also probed the ratio of PR-Ub in cells infected with the Δ*lnaB* mutant by mass spectrometric analysis, which revealed that more than 30% of ubiquitin was modified in the PR-Ub form under our experimental conditions (Fig. 4e). Finally, PR-Ub was not detected in cells infected with the Δ*dupA*Δ*dupB* mutant (Supplementary Fig. 9b), expression of either gene but not their enzymatically inactive mutants in this strain restored its accumulation in infected cells, indicating that PR-Ub is produced by reversal of SidEs-induced ubiquitination. Together, these results establish that LnaB functions to covert PR-Ub into ADPR-Ub in infected cells.

The conversion of PR-Ub and ADPR-Ub into native ubiquitin suggests that the accumulation of these ubiquitin derivatives interferes with the intracellular replication of *L. pneumophila*. We thus examined the growth of relevant *L. pneumophila* strains in *Dictyostelium discoideum*. Consistent with results from an earlier study[27], deletion of *lnaB* did not detectably impact intracellular bacterial replication (Fig. 4f). Overexpression of pSdeA in the wild-type strain

led to a reduction in bacterial growth, and such growth defects became more pronounced when SdeA was expressed in the Δ*lnaB* strain. In contrast, expression of the mART-defective mutant SdeA$_{E/A}$ or SdeA$_{H/A}$, the mutant defective in the phosphodiesterase (PDE) activity in the Δ*lnaB* mutant did not cause such defect (Fig. 4f). Importantly, the growth defect can be complemented by expressing LnaB but not its enzymatically inactive mutant (Fig. 4f). Taken together, these results suggest that accumulation of PR-Ub in host cells is detrimental to intracellular bacterial growth and the effects became more severe when the expression level of SidEs such as SdeA was increased in the bacterium, and that LnaB ameliorates such impact by eliminating it in infected cells.

## LnaB and tested members of the S-HxxxE family catalyze protein AMPylation

Incubation of $^{32}$P-α-ATP with LnaB in the absence of PR-Ub generated radio-labeled LnaB (Fig. 2f, **2nd lane**), suggesting that this enzyme is capable of catalyzing protein AMPylation by transferring the AMP moiety from ATP onto one or more of its own residues. Indeed, mass spectrometric analysis revealed that both Y196 and Y247 were AMPylated (Fig. 5a, b and Supplementary Fig. 9c, d). Mutations in both Y196 and Y247 gave rise to a mutant that had lost the ability to self-AMPylate but retained the activity to convert PR-Ub into ADPR-Ub (Fig. 5b, c). These results establish LnaB as an enzyme that catalyzes the cleavage of ATP at the α phosphate position to transfer the

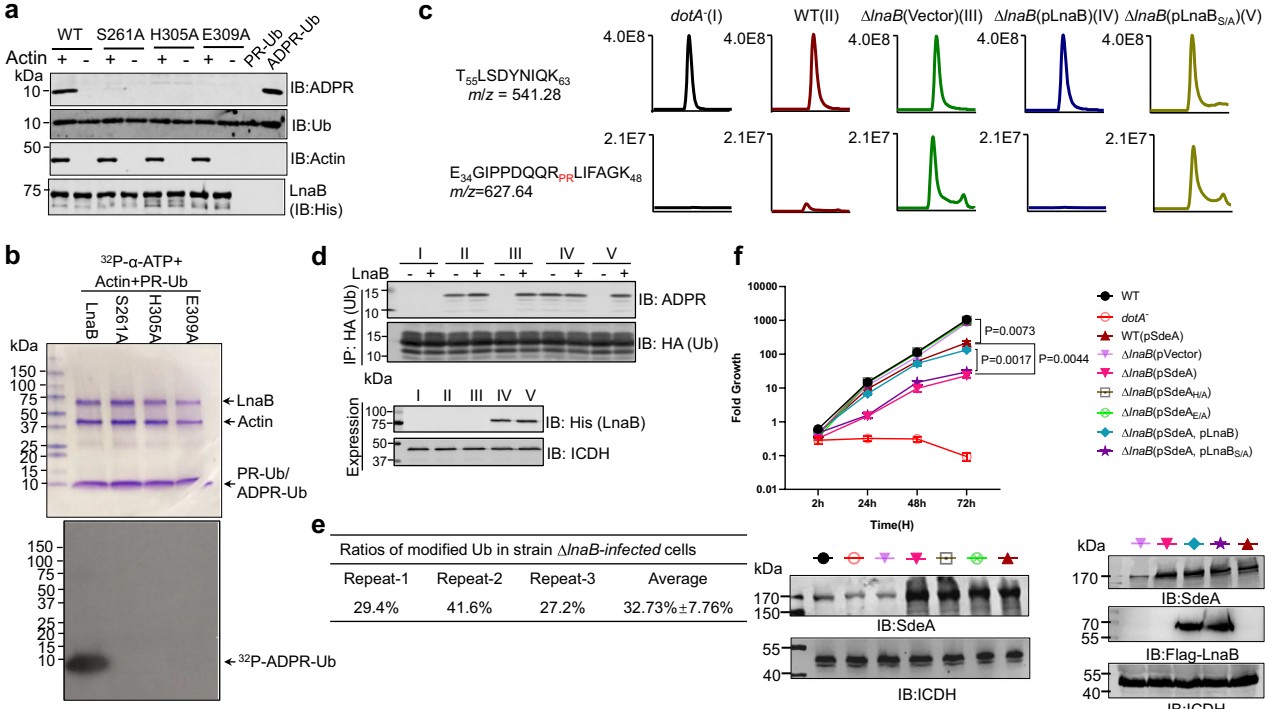

**Fig. 4 | The reaction catalyzed by LnaB required an S-HxxxE motif.**
**a**, **b** Conversion of PR-Ub into ADPR-Ub by LnaB requires an S-HxxxE motif. Samples of reactions containing ATP, PR-Ub, LnaB, Actin, LnaB, or its mutants and resolved by SDS-PAGE were detected for the production of ADPR-Ub (top). Each reactant was detected by immunoblotting with the appropriate antibodies (**a**). Similar reactions with $^{32}P$-$\alpha$-ATP were established, proteins were detected by CBB staining (upper), and the production of $^{32}P$-ADPR-Ub was detected by autoradiograph (lower) (**b**). **c**, **d** LnaB functions to convert PR-Ub into ADPR-Ub in cells infected with *L. pneumophila*. **c**, **d** HEK293 cells transfected to express 3xHA-Ub were infected with the indicated bacterial strains (I to V). Immunoprecipitation products obtained by HA antibody from lysates of infected cells were analyzed by mass spectrometry to detect differently modified ubiquitin (**c**). Recombinant LnaB was added to a subset of similarly prepared lysates of infected cells, and the accumulation of PR-Ub was

assessed by detecting LnaB-mediated ADPR-Ub production (**d**). **e** The ratio of modified ubiquitin (PR-Ub) in cells infected with the $\Delta lnaB$ mutant. Cells expressing HA-ubiquitin was infected with strain Lp02$\Delta lnaB$ for 2 h. HA-ubiquitin isolated by immunoprecipitation was analyzed by mass spectrometry to determine the ratio of modified ubiquitin. **f** Overexpression of SdeA in the $\Delta lnaB$ mutant affects intracellular bacterial growth *D. discoideum* was infected with the indicated *L. pneumophila* strains, and the growth of the bacteria was evaluated. Note that strain $\Delta lnaB$(pSdeA) displayed significant defects in intracellular growth (upper panel). The expression of SdeA in the testing strains was probed by immunoblotting (lower panel). The Data shown were representative of three independent experiments done in triplicate with similar results. Data are represented as mean ± SEM. P values were calculated using a two-sided Student *t* test.

---

AMP moiety onto the phosphoryl group in PR-Ub and the side chain of tyrosine residues.

To determine the AMPylator activity of the S-HxxxE family, we purified recombinant proteins of a selection of toxins and evaluated their self-modification using $^{32}P$-$\alpha$-ATP. For the five toxins examined, self-AMPylation was readily detected for the toxin from *Burkholderia ambifaria* (Tba675) and a fragment of the toxin from *Edwardsiella ictaluri* (Tei158) (Fig. 5d). Weak but detectable self-AMPylation was detected for WP_075066242.1 (0750) from *Candidatus Berkiella aquae*, WP_148338824.1 (1483) from *Aquicella siphonis*, MAZ44397.1 (MAZ443) from a bacterium of the Legionellales. In each case, an intact S-HxxxE motif was required for the activity as mutations in the serine residue abolished self-AMPylation (Fig. 5d). Furthermore, we found that Tei158 and Tba675 were toxic to yeast in a manner that requires an intact S-HxxxE motif (Fig. 5e). The lack of toxicity by other tested toxins suggested that their cellular targets are absent in yeast or that such targets are not essential for yeast viability.

**The LnaB-Actin complex reveals an AMPylator that recognizes ATP by a unique mechanism**

To analyze the catalytic mechanism of LnaB, we solved the crystal structure of the LnaB$_{19-371}$-Actin complex, which had activity indistinguishable from that of full-length protein (Fig. 6). The asymmetric unit (ASU) of the structure contains five LnaB$_{19-371}$ and six Actin molecules, which form three LnaB$_{19-371}$-Actin heterodimeric complex

(Supplementary Fig. 10a). In the LnaB$_{19-371}$-Actin complex, Actin mainly docks onto the carboxyl helix $\alpha14$ of LnaB$_{19-371}$ (Fig. 6a) with an interface area of 1763.9 Å². The structure of LnaB contains fourteen helixes and a long loop that folds into two subdomains: The N-terminal domain (NTD) and the catalytic domain (CD). A structural homology search of LnaB$_{19-371}$ with the DaLi server did not yield any significant hits, suggesting that it is a novel folding protein. Interestingly, S261, H305, and E309, the three residues critical for catalysis form a continuous platform located in an area that has concentrated positive electrostatic potential (Fig. 6b), which may be the site for protein-protein or protein-substrate interactions.

Two regions of LnaB$_{19-441}$ are in direct contact with Actin via extensive polar and hydrophobic interactions: a long loop consisting of a pair of antiparallel β-sheet proximal to the S-HxxxE motif and the carboxyl end helix $\alpha18$, which we designated as Interface1 and Interface2, respectively. In interface 1, T225 of LnaB engages by hydrogen-bonding interaction with K113 and R116 of Actin; N220 of LnaB forms hydrogen bonds with A170 and Y169 of Actin; T209 of LnaB contacts residues E286 of Actin via a hydrogen bond (Fig. 6c). In interface 2, H359 engages in hydrogen-bonding interaction with T148 and E167 of Actin; E370 and Q363 form hydrogen bonds with R147 of Actin (Fig. 6d). Other hydrogen bonds include R365(L, LnaB):S348(A, Actin), E361(L):T351(A), L352(L):Y169(A), D347(L):R372(A) and Q355(L):Y143(A). L362 of LnaB is inserted into a hydrophobic pocket composed of I345, L346, and Y134 of Actin. Substitution of T225, E361, or L362 with alanine indeed

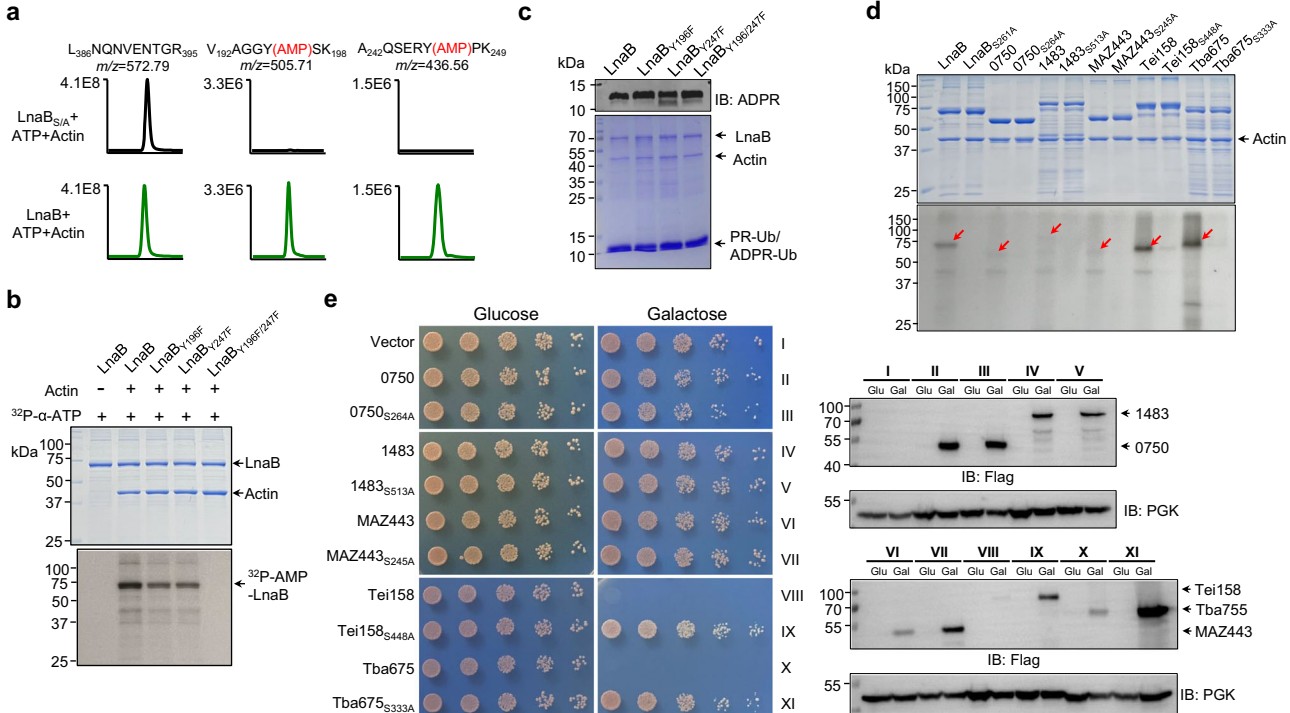

**Fig. 5 | Self-AMPylation activity of members of the S-HxxxE toxin family. a** LnaB self-AMPylates at Y196 and Y247. Protein bands corresponding to LnaB from the indicated reactions were analyzed to identify the modified residues by mass spectrometry. **b, c** Mutations of the AMPylated Tyr residues abolished the activity of LnaB. LnaB or its mutants was incubated with $^{32}$P-α-ATP and Actin and production of self-modified protein was detected by autoradiograph (**b**). Similar reactions receiving PR-Ub were established to probe the impact of the mutations on the conversion of PR-Ub into ADPR-Ub, which was detected by immunoblotting (top), and the proteins in the reactions were detected by CBB staining (lower). **d** Self-AMPylation by members of the S-HxxxE family. Recombinant proteins of the indicated toxins were incubated with $^{32}$P-α-ATP and Actin. Samples resolved by SDS-PAGE were detected for AMPylation by autoradiograph (lower) and for the proteins by CBB staining (upper). Red arrows indicated AMPylated proteins. Note that in each case, self-AMPylation required an intact S-HxxxE motif. **e** Yeast toxicity by the toxins required an intact S-H$_{XXX}$E motif. Serially diluted cells of yeast strains expressing the indicated toxin genes or their S-HxxxE mutants were spotted on a medium containing glucose or galactose. Images were acquired after 3-day incubation at 30 ℃ (left). The expression of the proteins was probed by immunoblotting with the Flag-specific antibody. The phosphoglycerate kinase (PGK) was probed as a loading control.

---

reduced LnaB activity toward PR-Ub. Yet, single substitution mutation in other sites involved in its interaction with Actin did not detectably affect its enzymatic activity (Fig. 6e, f).

Despite extensive efforts, we were unable to obtain crystals of the LnaB-Actin complex containing ATP. We thus docked the ATP molecule into LnaB by molecular docking, which revealed that ATP may bind to a positively charged pocket neighboring the S261-H305-E309 motif of LnaB (Fig. 6g and Supplementary Fig. 10b). Alanine substitution in K199, Y299 or Y304 markedly reduced LnaB activity (Fig. 6h) and the binding of these mutants to ATP became almost undetectable (Fig. 6i and Supplementary Fig. 11), validating our hypothesis that this positively charged pocket is involved in ATP binding.

## Discussion

Three mechanisms for protein AMPylation have been described[28], including the Cx$_{11}$DxD motif employed by the glutamine synthetase from *E. coli* and the multifunctional *L. pneumophila* effector SidM/DrrA[29–31], the Fic domain exemplified by VopS of *Vibrio parahaemolyticus*[32] and the pseudokinase domain found in widely distributed proteins of the selenoprotein-O protein family[33]. Differing from enzymes that only target hydroxyl-containing side chains of their substrate, LnaB targets the phosphate group in PR-Ub. Self-AMPylation by members of the S-HxxxE family suggests that at least a fraction of these enzymes AMPylate their substrate in host cells. The Fic domain has been shown to use CDP-choline[34] and UTP[35] as reactants, it is possible that some S-HxxxE proteins may use other nucleotides or their derivatives as substrates for protein modification. AMPylation is a reversible modification, signaling by this mechanism can be modulated by specific stimuli[36–38]. It is of great interest to identify enzymes involved in the reversal of AMPylation catalyzed by these diverse modifiers.

The requirement of a host cell-specific co-factor allows pathogens to restrict the activity of virulence factors within target cells. For instance, both the edema factor of *Bacillus anthracis* and CyaA of *Bordetella pertussis*[39,40] use calmodulin (CaM) as the co-factor to ensure that cAMP is generated only in host cells. The requirement of CaM by the glutamylase SidJ is to prevent premature inactivation of SidEs in *L. pneumophila*[20,21]. PR-Ub has only been found in cells infected by *L. pneumophila*[7,9,10] produced by DupA and DupB from proteins modified by SidEs. Our results have highlighted the importance of ubiquitin homeostasis in cells infected by *L. pneumophila* (Fig. 7). Yet, the reason for the requirement of Actin for LnaB activity is less clear. Actin dependence may prevent LnaB-induced ATP depletion, or active LnaB may recognize phosphoribosyl-bearing metabolites in bacterial cells. Alternatively, binding to Actin may facilitate the targeting of host proteins of relevant cellular processes such as NFκB signaling. Further investigation of the catalytic mechanism of LnaB, the function of toxins of this family, and the potential use of the S-HxxxE motif by eukaryotic cells in signaling will shed insights into not only protein biochemistry but also novel cell signaling cascades potentially important for development and disease.

## Methods

### Media, bacteria strains, plasmid construction and cell lines

*Escherichia coli* strains were grown on LB agar plates or in LB broth. When necessary, antibiotics were added to media at the following

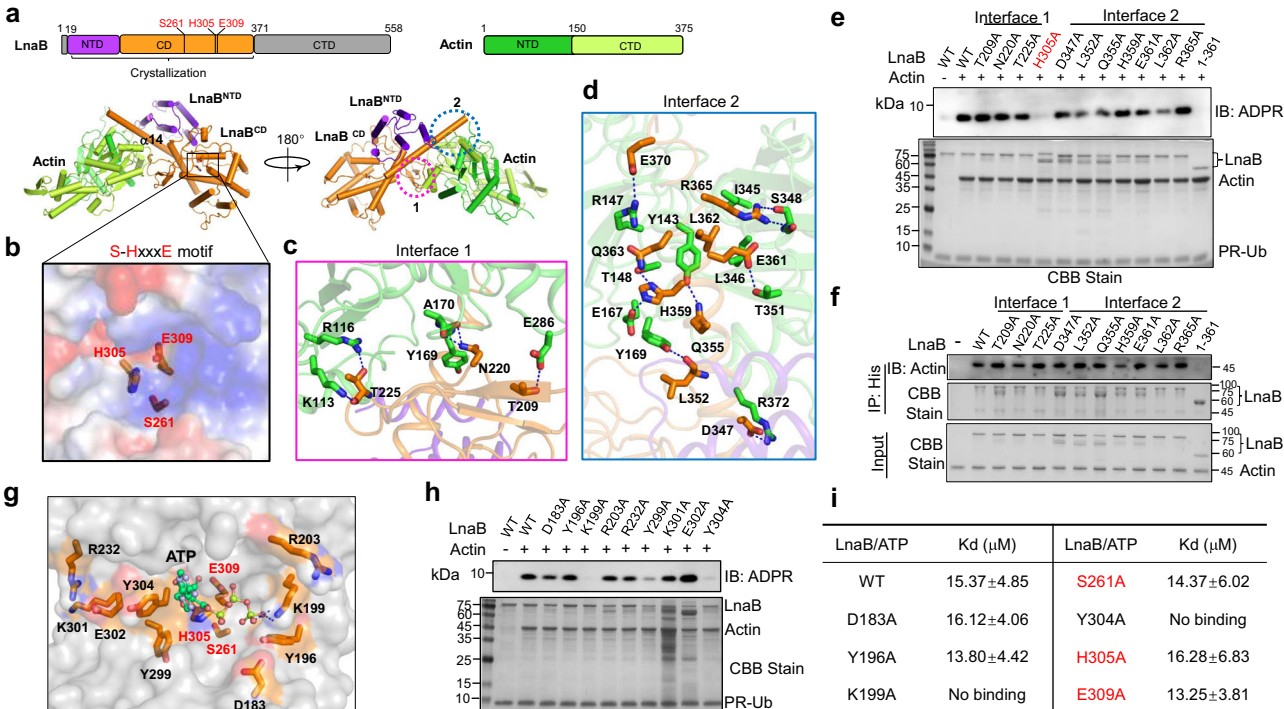

**Fig. 6 | LnaB-Actin binary complex structure reveals a unique catalytic mechanism on AMPylation. a** Cylindrical cartoon diagram representation of the LnaB-Actin complex. The top panels represent schematic diagrams of the regions for domain organization of LnaB and Actin. LnaB consists of the N-terminal domain (NTD, purple), the catalytic domain (CD, orange), and the C-terminal domain (CTD, gray); Actin is composed of NTD (green) and CTD (limon). S261, H305, and E309 of the S-HxxxE motif are shown in red. The bottom panel shows the LnaB-Actin binary structure. Domains of LnaB and Actin were colored in accordance with the diagrams (top). The interfaces involved in LnaB-Actin interactions were highlighted in two dashed line circles. **b** S261, H305, and E309 formed a platform in the structure of LnaB. Residues were represented as sticks and LnaB was depicted in surface, colored according to the electrostatic surface potential [contoured from − 6kBT (red) to + 6kBT (blue)]. **c, d** The interfaces involved in LnaB-Actin interactions. LnaB and Actin were shown as orange and green cartoons, respectively. Residues important for binding were shown as sticks (Actin in green and LnaB in orange). Hydrogen bonds were marked by blue dashed lines. **e** Optimal binding to Actin is required for maximal activity of LnaB. Indicated LnaB mutants were individually

incubated with Actin, ATP, and PR-Ub for 30 min at 37 °C and their activity in converting PR-Ub into ADPR-Ub was evaluated by immunoblotting with an ADPR-specific antibody. Proteins in the reactions were detected by CBB staining. **f** Evaluation of the binding of Actin to LnaB and its mutants by Ni²⁺ beads pulldown. His₆-LnaB and its mutants were individually incubated with Actin at 4 °C for 6 h prior to pulldown with Ni²⁺ beads. Actin was detected using anti-Actin antibodies and proteins were detected by CBB staining. **g–i** An ATP-binding pocket in LnaB identified by molecular docking. LnaB was displayed in a gray surface model. Residues potentially involved in binding ATP was indicated as orange sticks. ATP was shown as a cyan stick-ball model and hydrogen bonds were represented by blue dashed lines (**g**). LnaB mutants were evaluated for the ability to convert PR-Ub into ADPR-Ub with the reactions described above. Proteins were detected by CBB staining (**h**). The affinity between ATP and LnaB and its mutants was determined using isothermal titration calorimetry (ITC). The binding constant (Kd) was calculated by the NanoAnayze software package. The Data shown are representative of three independent experiments with similar results (**e**, **f**, **h**, and **i**).

concentrations: ampicillin, 100 µg/mL; kanamycin, 30 µg/mL. *L. pneumophila* strains used in this study were derivatives of the Philadelphia 1 strain Lp02[41]. Lp03 is an isogenic *dotA⁻* mutant[41]. All strains were grown and maintained on CYE plates or in ACES-buffered yeast extract (AYE) broth as previously described[41]. When needed, thymidine was added at a final concentration of 100 µg/mL. The Lp02Δ*dupA*Δ-*dupB* mutant was described earlier[9]. Mutants lacking *mavL* or *lnaB* were constructed using strain Lp02 as previously described[42]. Complementation plasmids were constructed by inserting the gene of interest into pZL507[43]. The plasmid for expression SdeA, mutants SdeA_E/A and SdeA_H/A were from previous studies[6,20]. For ectopic expression of proteins in mammalian cells, genes were inserted into pEGFPC1 (Clontech) or p4xFlagCMV[6]. The plasmid for expressing 3xHA-Ub in mammalian cells had been described earlier[6]. Genes for purifications were cloned into pGEX-6P-1 (Amersham), pQE30 (QIAN-GEN), pET-28a (Novagen) or pET28a-Sumo (Novagen). The integrity of all constructs was verified by sequencing analysis. Genes coding for WP_015869719.1 from *Edwardsiella ictaluri*, WP_006755655.1 from *Burkholderia ambifaria,* and WP_075066242.1 from *Candidatus Berkiella aquae*, WP_148338824.1 from *Aquicella siphonis*, MAZ44397.1 from a species of Legionellales were synthesized by GenScript Biotech

Corp (Nanjing, China) with codon optimized for *E. coli*. HEK293T cells purchased from the ATCC were cultured in Dulbecco's modified minimal Eagle's medium (DMEM) supplemented with 10% Fetal Bovine Serum (FBS). All mammalian cell lines were regularly checked for potential mycoplasma contamination by the universal mycoplasma detection kit from ATCC (Cat# 30-1012 K).

## Yeast toxicity assays

Yeast strains were grown in YPD (1% yeast extract, 2% peptone, 2% glucose) or SD minimal media containing nitrogen base, glucose, and amino acid drop-out mix for selection of transformed plasmids as described[44]. The genes coding for the testing toxins was individually inserted into pYES2/NTA (Invitrogen) that carries a galactose-inducible promoter[45]. In each case, the sequence coding for the Flag tag was added to the amino terminal end of the gene to facilitate the detection of gene expression. The resulting plasmids were introduced into yeast strain W303[46], respectively. Ten microliters of 5-fold dilutions of saturated cultures were spotted in a dropout medium containing glucose or galactose. Plates were incubated at 30 °C for 3 d prior to image acquisition to assess growth. To detect protein expression, cells cultured in medium containing 2% raffinose were washed once with

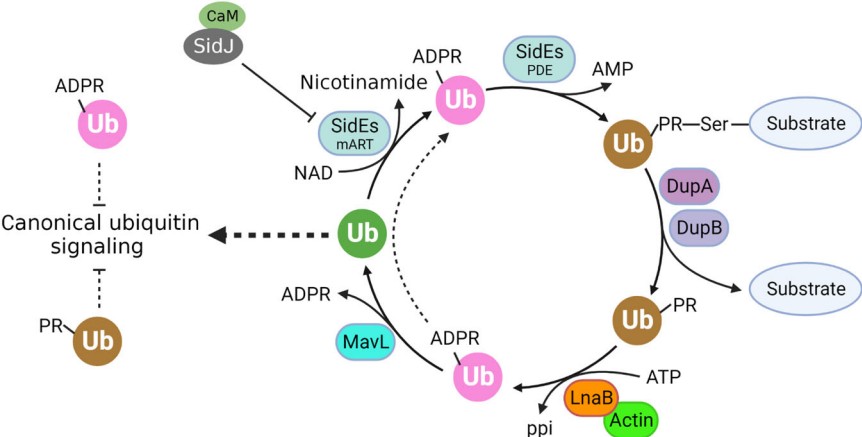

**Fig. 7 | The cycling of ubiquitin by Dot/Icm effectors in cells infected by _L. pneumophila._** Ubiquitin is converted into ADPR-Ub by the mART activity of SidEs, which is used to modify proteins by phosphoribosyl ubiquitination. The reversal of the modification produced PR-Ub, which is converted into native ubiquitin by sequential reactions catalyzed by LnaB and MavL. Note that both ADPR-Ub and PR-Ub may interfere with canonical ubiquitin signaling and that ADPR-Ub produced from PR-Ub by LnaB may be used by the PDE activity of SidEs for protein modification.

galactose medium and were induced in 2% galactose medium for 2 d at 30 °C.

### Transfection, infection and immunoprecipitation

Plasmids were transfected into mammalian cells by using Lipofectamine 3000 (Invitrogen, cat# L3000150). After 24 h transfection, cells were collected and lysed with the TBS buffer (150 mM NaCl, 50 mM Tris-HCl, pH 7.5) with 1% Triton X-100. When needed, immunoprecipitation was performed with lysates of transfected cells by Flag-specific antibody-coated agarose beads (Sigma, cat# F2426) or GFP-specific antibody, which was coupled to protein G beads (Cytiva, cat# 17061801) at 4 °C for 8 h. Beads were washed 3x with pre-cold lysis buffer. After that, Flag peptide solution (150 μg/mL) (Sigma, cat# F3290) or 0.1 M glycine buffer (pH 2.5) was used to elute Flag-tagged proteins or GFP-tagged proteins, respectively. All samples were resolved by SDS-PAGE and followed by immunoblotting analysis with the specific antibodies.

For intracellular bacterial growth, _L. pneumophila_ strains grown to early post-exponential phase ($OD_{600} = 3.3$–3.8) were used to infect bone marrow-derived macrophages or _D. discoideum_ at an MOI of 0.05 as described earlier[42]. 2 h after adding the bacteria, infections were synchronized by washing cells with warm PBS to remove the extracellular bacteria, and IPTG was added at a final concentration of 0.05 mM to induce the expression of SdeA and its mutant. Infected cells were lysed with 0.02% saponin at the indicated time points, and the total viable bacterial cells were determined by plating appropriate dilutions on CYE plates.

To determine ADPR-Ub levels in infected cells, HEK293T cells were co-transfected with plasmids expressing the FcγII receptor and 3xHA-Ub[6]. The indicated _L. pneumophila_ strains were grown to the post-exponential growth phase ($OD_{600} = 3.4$–3.8) in AYET broth containing Kanamycin (20 μg/mL). Four h prior to infection, IPTG was added into the broth at a final concentration of 0.2 mM. _L. pneumophila_ cells were opsonized by mixing with _L. pneumophila_-specific rabbit antibodies at a 1:500 ratio for 30 min at 37 °C. Thirty-six h post-transfection, opsonized bacteria were used to infect transfected cells at an MOI of 50. 2 h post-infection, cells were washed with cold PBS and lysed with TBS buffer containing 150 mM NaCl, 50 mM Tris-HCl, 1 mM DTT, 1% Triton X-100, following with 10% sonication for 10 s. Lysates were centrifuged twice at $20,000 \times g$ for 15 min, and the supernatants were collected and incubated anti-HA beads at 4 °C for 8 h. Beads were washed three times with cold lysis buffer and were boiled in 1x sample buffer for 10 min.

To detect PR-Ub in cells infected with _L. pneumophila_ strains, cell lysates obtained using an injection procedure described above were divided into two identical samples, and 10 μg His$_6$-LnaB was added to one sample, and 10 μL TBS buffer was added to another sample as a control. The reactions were allowed to proceed for 2 h at 37 °C and samples were then incubated with anti-HA beads to immunoprecipitated 3xHA-Ub by incubation at 4 °C for 8 h. Beads were washed three times in cold lysis buffer and were boiled in 1x sample buffer for 10 min prior to SDS-PAGE.

To determine the association of MavL and LnaB with the LCV, HEK293 cells transfected to express HA-MavL or 4Flag-LnaB were infected with the relevant _L. pneumophila_ strains for 1 h, and the samples were stained with the appropriate antibodies. The intracellular and total bacteria were distinguished by sequential immunostaining. Samples were inspected and analyzed using an Olympus IX-83 florescence microscope.

### Protein purification

For His$_6$- or GST-tagged recombinant protein production for in vitro assays, 20 mL saturated _E. coli_ cultures were transferred to 400 mL LB medium supplemented with 30 μg/mL kanamycin or 100 μg/mL ampicillin, the cultures were grown to $OD_{600nm}$ of 0.6–0.8 at 37 °C. Protein expression was induced with 0.2 mM IPTG at 18 °C for 16–18 h on a shaker (200 rpm). Bacterial cells were collected by centrifugation and lysed by sonication. The soluble lysates were cleared by spinning at $15,000 \times g$ at 4 °C for 30 min. Supernatant containing recombinant proteins were purified by Ni$^{2+}$-NTA beads (QIAGEN) or Glutathione agarose beads (Pierce), and were eluted from beads by using PBS buffer containing 300 mM imidazole or 10 mM reduced glutathione in a Tris buffer (50 mM Tris-HCl, pH 8.0). Purified proteins were dialyzed against PBS buffer containing 10% glycerol and 1 mM DTT at 4 °C for 8 h.

To purify proteins for structural study, the coding regions or the truncation mutants of MavL or LnaB were inserted into pET28a-sumo and the resulting plasmids each were transformed into _E. coli_ strain BL21(DE3). The bacterial strains were cultured at 37 °C in LB broth on a shaker (220 rpm) and then induced with 0.5 mM IPTG when the bacteria grew to a density OD600 = 0.8. The bacteria were then cultured for 16 h at 18 °C before collecting the cells by centrifugation ($5000 \times g$, 15 min). Cells resuspended in a cold lysis buffer (50 mM Tris–HCl pH 7.5, 150 mM NaCl) were lysed by ultrasonication. Lysates were centrifuged at $35,000 \times g$ for 30 min at 4 °C to obtain supernatant, which was used to purify His$_6$-tagged proteins by affinity chromatography

(Ni$^{2+}$ resin). The SUMO tag was removed by the SUMO protease ULP1. Proteins were further purified by size-exclusion chromatography using a Superdex 200 Increase column (GE Healthcare) equilibrated with a buffer containing 25 mM HEPES, pH 7.5, 150 mM NaCl, and 2 mM DTT. The best fractions of protein peak were pooled, concentrated to 10 mg/mL with an Amicon Centrifugal filter (Millipore), flash-frozen in liquid nitrogen, and stored at −80 °C for crystallization and activity assay. Protein concentration was measured at A280 and calculated using their theoretical extinction coefficients.

## Isothermal titration calorimetry

Isothermal titration calorimetry ITC experiments were carried out using a Low Volume Nano ITC (TA instrument) set at 20 °C. The protein of MavL or its mutants and ADPR were dissolved in the buffer containing 50 mM Tris-HCl, pH 7.5, and 150 mM NaCl. The concentration of ADPR in the syringe was 1 mM and the concentrations of proteins in the sample cell were 0.1 mM. Twenty-five consecutive 2 μl injections of ADPR were titrated into a 350 μl sample cell with a 200 s interval between injections using a stirring rate of 120 rpm. A single-site binding model was used for nonlinear curve fitting using the Launch NanoAnalyze software provided by the manufacturer.

LnaB-Actin binding was performed using a microcalorimeter Affinity ITC (Waters$^{TM}$, USA) at 20 °C. LnaB and its mutants purified as above was diluted into a buffer of 50 mM Tris-HCl and 150 mM NaCl to a final concentration of 0.2 mM. Actin (Sangon Biotech, A001041) was prepared in the same buffer at a concentration of 22 μM. To measure LanB-ATP binding, Concentrations of LnaB or its mutants and ATP were 0.1 mM and 1 mM, respectively. Titrations were set for 20 injections each of 2 μL with 200 s intervals except from the first injection of 0.2 μL. Baseline subtraction and data analysis were performed using NanoAnayze (Waters$^{TM}$, USA). Heat spikes were integrated and fitted with a 1:1 binding model. The first injection was excluded from the analysis.

## Analytical ultracentrifugation (AUC)

Sedimentation velocity (SV) experiments were performed in a buffer containing 50 mM Tris-HCl, pH 8.0, and 150 mM NaCl using Proteomelab XL-I Analytical Ultracentrifuge (Beckman-Coulter). In AUC-SV analysis, runs were carried out at 50,000 r.p.m. and at a temperature of 10.0 °C using 12 mm charcoal-epon double sector centerpieces and An60 Ti analytical rotor. The evolution of the resulting concentration gradient was monitored with absorbance detection optics at 280 nm, All AUC-SV raw data were analyzed by the continuous C(s) distribution model implemented in the program SEDFIT[47]. Partial specific volume and extinction coefficient of the protein as well as buffer density and viscosity, were calculated from amino acid and buffer composition, respectively, by the program SEDNTERP[48] and and were used to calculate protein concentration and correct experimental s-values to s$_{20,w}$.

## Ni$^{2+}$-agarose affinity pull-down assays

This assay was carried out in the binding buffer 150 mM NaCl, 20 mM Tris pH 8.0, 20 mM imidazole, and 0.02% Triton X-100. 1 mL of the reaction mixture, including 70 μg His-tagged LnaB or its mutants and 20 μg Actin, was incubated at 4 °C. After 4 h, 20 μL of nickel sepharose beads were added and incubated for another 1 h. The bound beads were added with 1.25 × SDS-loading buffer and boiled at 95 °C for 10 min after washing three times with the binding buffer, then separating by SDS-PAGE and detecting by an anti-Actin antibody.

## Biochemical AMPylation assays

In a 20 μL reaction, 1 μg His$_6$-LnaB or its mutants, 1 μg Actin (Cytoskeleton, cat# APHL99), and 1.5 μg PR-Ub were used in a solution containing 50 mM Tris-HCl (pH 7.5), 5 mM MgCl$_2$ and 1 mM ATP, and the reaction was allowed to proceed for 1 h at 37 °C. To measure the

activity of LnaB using ATP-α-$^{32}$P, 1 μg His$_6$-LnaB, 1 μg Actin (Cytoskeleton, cat# APHL99) and 1.5 μg PR-Ub were incubated in a 20 μL reaction system containing 50 mM Tris-HCl (pH 7.5), 5 mM MgCl$_2$ and 5 μCi ATP-α-$^{32}$P (Perkin Elmer, cat# BLU003H250UC) for 1 h at 37 °C. Samples were resolved by SDS-PAGE, and gels were stained with Coomassie brilliant blue. Gels were then dried and the signals were detected with x-ray films.

For the AMPylation activity of the selected members of the S-HxxxE family proteins, 1 μg His$_6$-LnaB or homologous proteins and 0.5 μg actin were incubated in a 20 μL reaction system containing 50 mM Tris-HCl (pH 7.5), 5 mM MgCl$_2$ and 5 μCi ATP-α-$^{32}$P for 2 h at 37 °C. Products were resolved by 12% SDS-PAGE at 100 V for 2 hr. Gels were stained with Coomassie brilliant blue (CBB) for 1 h, de-stained twice for 2 h and then dried for 8 h. Signals were detected with X-ray films using a BioMax TranScreen LE (Kodak) for 4 h at RT.

## Biochemical de-ADP-ribosylation assays

To examine the activity of MavL on different modified ubiquitin, 2 μM MavL was incubated with 100 μM ADPR-Ub, PR-Ub or Ub for 30 min at 37 °C in a solution containing 50 mM Tris-HCl (pH 7.5). Proteins in reactions resolved by native PAGE or SDS-PAGE were detected by Coomassie blue stain or by immunoblotting with the anti-ADPR antibody. To determine the activity of MavL, DupA, and DupB against ADPR-Ub, 2 μM MavL, DupA, or DupB was incubated with 100 μM ADPR-Ub for 30 min at 37 °C in a solution containing 50 mM Tris-HCl (pH 7.5). Samples resolved by native PAGE or SDS-PAGE were detected by Coomassie blue stain, phosphoprotein stain (ABP Biosciences), or immunoblotting with the anti-ADPR antibody.

To examine the specificity of the de-ADP-ribosylation activity of MavL, we prepared several ADP-ribosylated proteins including ADPR-Actin catalyzed by SpvB of *Salmonella enterica*[14], ADPR-ANT1 by Ceg3 of *L. pneumophila*[15], ADPR-PARP1 by Sirt6[16]. In each case, the substrate protein and the enzyme were co-expressed in 293HEK cells by transfection, and the ADP-ribosylated proteins were isolated by immunoprecipitation using beads coated with antibody specific for the Flag or HA tag. ADPR-T66-Ub was generated by incubating His$_6$-ubiquitin with GST-CteC of *C. violaceum*[13]. Each of the ADP-ribosylated proteins was incubated with MavL or MavL$_{D323A}$ for 30 min at 37 °C and de-ADP-ribosylation effects were probed by immunoblotting using the anti-ADPR antibody.

## Biochemical ubiquitination assays

For the SdeA-mediated ubiquitination reaction, 2 μg His$_6$-LnaB, 1.5 μg Actin (Cytoskeleton, cat# APHL99) and 6 μg PR-Ub were preincubated in a 25 μL reaction system containing 50 mM Tris-HCl (pH 7.5), 5 mM MgCl$_2$ and 1 mM ATP for 1 h at 37 °C. After preincubation, a cocktail containing 0.1 μg His$_6$-SdeA$_{E/A}$ and 0.6 μg His$_6$−4xFlag-Rab33b were supplemented into reactions, and the reaction was allowed to proceed for another 2 h at 37 °C.

For SidC-mediated ubiquitination reaction[49], 0.3 μg GST-E1, 1 μg His$_6$-UbcH7, 3 μg GST-SidC$_{1-542}$ and 4 μg Ub were incubated in a 20 μL reaction system containing 50 mM Tris-HCl (pH 7.5), 5 mM MgCl$_2$, 1 mM DTT and 2 mM ATP for 1 h at 37 °C.

To test the ability of LnaB and MavL to convert PR-Ub into active ubiquitin, 1.5 μg His$_6$-LnaB, 1.5 μg Actin, 1.5 μg GST-MavL and 4 μg PR-Ub were preincubated in a 25 μL reaction system containing 50 mM Tris-HCl (pH 7.5), 5 mM MgCl$_2$ and 1 mM ATP for 1 h at 37 °C. After preincubation, reactions were boiled for 5 min at 95 °C, then a cocktail containing 0.3 μg GST-E1, 1 μg His$_6$-UbcH7, 3 μg GST-SidC$_{1-542}$, 1 mM DTT, and 2 mM ATP were supplemented into these boiled reactions and the reaction was allowed to proceed for another 1 h at 37 °C.

## Antibodies and immunoblotting

Purified His$_6$-GFP and His$_6$-GST proteins were used to raise rabbit-specific antibodies using a standard protocol (Pocono Rabbit Farm &

Laboratory). These antibodies were affinity purified as described. Antibodies specific for SdeA had been described[6]. For immunoblotting, samples resolved by SDS-PAGE were transferred onto 0.2 μm nitrocellulose membranes (Bio-Rad, cat# 1620112), which were blocked with 5% non-fat milk or 3% BSA at R.T. for 1 h prior to being incubated with the appropriate primary antibodies: anti-Flag (Sigma, cat# F1804), 1:5000; anti-Actin (MP Biochemicals, cat# 0869100), 1:5000; anti-tubulin (DSHB, E7) 1:10,000; anti-ADPR (Sigma, cat# MABE1016), 1:1000, anti-His (Sigma, cat# H1029), 1:5000; anti-Ub (Santa Cruz, P4D1, cat# sc-8017), 1:1000. Membranes were then incubated with appropriate IRDye infrared secondary antibodies and scanned by an Odyssey infrared imaging system (Li-Cor's Biosciences).

## LC-MS/MS analysis

Protein bands were digested in-gel with trypsin as previously described[50]. Digested peptides were analyzed by LC-ESI-MS/MS using the Dionex UltiMate 3000 RSLC nano System coupled to the Q Exactive™ HF Hybrid Quadrupole-Orbitrap Mass Spectrometer (Thermo Scientific, Waltham, MA). The reverse phase peptide separation was accomplished using a trap column (300 μm ID × 5 mm) packed with 5 μm 100 Å PepMap C18 medium, and then separated on a reverse phase column (50 cm long × 75 μm ID) packed with 2 μm 100 Å PepMap C18 silica (Thermo Fisher Scientific, Waltham, MA). The column temperature was maintained at 50 °C.

Mobile phase solvent A was 0.1% FA in water, and solvent B was 0.1% FA in 80% ACN. Loading buffer was 98% water/2% ACN/0.1% FA. Peptides were separated by loading into the trap column in a loading buffer for 5 min at a 5 μL/min flow rate and eluted from the analytical column at a flow rate of 150 μL/min using a 130 min LC gradient as follows: linear gradient of 5% to 27% of solvent B in 80 min, 27–45% in next 20 min, 45–100% of B in next 5 min at which point the gradient was held at 100% of B for 7 min before reverting back to 2% of B at 112 min, and held at 2% of B for next 18 min for equilibration. The mass spectrometer was operated in positive ion and standard data-dependent acquisition mode with the Advanced Peak Detection function activated for the top 20n. The fragmentation of precursor ions was accomplished by a stepped normalized collision energy setting of 27%. The resolution of the Orbitrap mass analyzer was set to 120,000 and 15,000 for MS1 and MS2, respectively. The full scan MS1 spectra were collected in the mass range of 350–1600 $m/z$, with an isolation window of 1.2 $m/z$ and a fixed first mass of 100 $m/z$ for MS2. The spray voltage was set at 2, and the Automatic Gain Control (AGC) target of 4e5 for MS1 and 5e4 for MS2, respectively.

For protein identification, the raw data were processed with the software MaxQuant (version 1.6.3.3) against the *Homo sapiens* database (Uniprot, UP000005640) or *L. pneumophila* database (Uniprot, UP000000609). MaxQuant was set to search with the following parameters: peptide tolerance at 10 ppm, MS/MS tolerance at 0.02 Da, carbamidomethyl (C) as a fixed modification, oxidation (M) as a variable modification, and a maximum of two missed cleavages. The false-discovery rates (FDR) were controlled at <1%. To identify the Phosphoribosylation or ADP-ribosylation modification peptides, raw data were analyzed manually in Xcalibur QualBrowser.

## Crystallization, data collection, and structural determination

Crystallization of the complexes of MavL-ADPR, MavL$_{40-404(D315A)}$-ADPR-Ub, and LnaB-Actin were conducted using the hanging-drop vapor diffusion method at 16 °C, with drops containing 0.5 μl of the protein solution mixed with 0.5 μl of reservoir solution. Diffraction quality of the MavL-ADPR complex crystals was obtained in 0.25 M potassium citrate tribasic monohydrate,18% PEG3350, and MavL$_{40-404(D315A)}$-ADPR-Ub complex crystals was obtained in 0.1 M HEPES/sodium hydroxide pH7.5, 20% polyethylene glycol 10,000, respectively. LnaB-Actin complex crystals were observed in 0.15 M Ammonium sulfate, 0.1 M Sodium HEPES pH 7.0, 20% (w/v) polyethylene glycol (PEG) 4000 after one week. After optimization, the best crystals of LnaB-Actin were obtained in 0.15 M Ammonium sulfate, 0.1 M Sodium HEPES pH 7.0, 22% (w/v) PEG 4000. Crystals were harvested and flash-frozen in liquid nitrogen with 20% glycerol as a cryoprotectant. Complete X-ray diffraction data sets were collected at the BL02U1 beamline of the Shanghai Synchrotron Radiation Facility (SSRF). Diffraction images were processed with the HKL-2000 program. Molecular Replacement was then performed with the model of Apo MavL (PDB:6OMI) and Ub (PDB:6K11) as a template to determine the structure of the MavL-ADPR (PDB:8IPW) and MavL$_{40-404(D315A)}$-ADPR-Ub (PDB:8IPJ) complexes, respectively. The Actin and LnaB structures (1–361 region) predicted with Alphafold2[51] were employed as a template. Model building and crystallographic refinement were carried out in Coot and PHENIX[52]. Detailed data collection and refinement statistics are listed in Supplementary Table 1. The interactions were analyzed with PyMOL (http://www.pymol.org/), and PDBsum and figures were generated with PyMOL.

## Bioinformatic identification of members of the S-HxxxE family

To obtain LnaB orthologous sequences from other genera (we excluded the *Legionella* genus from the blast search), we utilized PSI-BLAST[25] (Position-Specific Iterative Basic Local Alignment Search Tool) searches at the NCBI (National Center for Biotechnology Information; http://blast.ncbi.nlm.nih.gov/) against the nr (non-redundant) protein database. We limited the PSI-BLAST search to three rounds to minimize the effect of possible convergent evolution while still being able to detect all the genera that contain LnaB orthologs. For the purpose of phylogenetic analysis and tree construction, we first removed duplicate sequences from the same species and selected closely related protein sequences for analysis. Sequence alignment was generated by MAFFT[53], and the resulting alignment was used to infer a phylogenetic tree using IQ-TREE[54]. VT + F + R4 model was selected by the ModelFinder[55] with 1000 nonparametric replicated bootstrap analyses. The obtained phylogenetic tree was visualized with iTOL web-server[56].

## Molecular docking

The configuration of ATP was optimized at the B3LYP/6-31 G* level using the Gaussian 09 package[57]. The structure of LnaB was extracted from the crystal structure of the LnaB/Actin complex resolved in this work. We carried out 300 independent docking runs with the Auto-Dock 4.2 program[58] using the Lamarckian genetic algorithm (LGA)[59] as a search engine. The size of the grid box was 90 × 90 × 90 with a grid spacing of 0.375 Å. The grid center was set at the center of mass of LnaB. ATP was treated as a flexible molecule whereas LnaB was rigid. The results of 300 docking runs were grouped into clusters according to the ligand binding conformation with default parameters implemented in AutoDockTools 1.5[58]. Among these cluster, there was one dominant cluster accounting for 68.7% of total docking conformations (Supplementary Fig. 10b). Furthermore, this cluster held the lowest mean binding energy (− 6.53 ± 0.41 kcal/mol) as calculated by Auto-DockTools 1.5 (Supplementary Fig.10b), indicating a robust structural stability. Thus, the conformation with the lowest binding energy from this cluster was selected as the putative complex model. The residues in the ATP binding site were defined by an atomic distance-based cutoff: the residues contained at least one atom within 4.5 Å of any atom in the bound ATP. Accordingly, the binding site residues of ATP included residues Y196, K199, R201, P235, I237, A258-G262, and G303-H305.

## Statistics and reproducibility

The Student's *t* test was used to compare the mean levels between two groups, each with at least three independent samples. All immuno-blotting results shown are representative of three independent experiments.

**Reporting summary**

Further information on research design is available in the Nature Portfolio Reporting Summary linked to this article.

## Data availability

Structure factors and atomic coordinates have been deposited in the Protein Data Bank (PDB) under accessions: MavL-ADPR (PDB:8IPW), MavL40-404(D315A)-ADPR-Ub (PDB:8IPJ), LnaB-actin (PDB:8J9B). The data that support the conclusions of this study are included in this published article along with its Supplementary Information files and are also available from the corresponding author upon request. LnaB interacting proteins identified by IP-MS are included in Supplementary Dataset 1, including full gels and blots provided in the Source Data file. Source data are provided as a Source Data file. Source data are provided with this paper.

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

## Acknowledgements

This work was supported in part by the National Key Research and Development Program of China 2021YFC2301403 (S.O.) and 2022YFA1304500 (X.L.), the National Natural Science Foundation of China grants 32370185 (J.F.), 32270185 (L.S.), 82225028 and 82172287 (S.O.), 31900879 and 32171265 (H.G.), 21974002 and 22174003 (X.L.), and the High-level personnel introduction grant of Fujian Normal University (Z0210509) (S.O.), and by National Institutes of Health grants R01AI127465 and R01GM126296. Mass spectrometric analysis was performed in the core facility of the First Hospital of Jilin University, the authors thank Dr. Naicui Zhai for assistance.

## Author contributions

Z.Q.L., J.F., X.L., S.O., and L.S. conceived the ideas for this work. Unless otherwise specified, J.F., S.L., and C.L. performed the biochemical experiments and infection experiments. L.S. performed the yeast experiments; H.G. and Z.Z. performed the experiments with MavL, Y.L., and Q.G. performed the bioinformatics analysis, J.F., W.X., and X.L. performed mass spectrometric analyses. H.G., Y.Z., T.T.C., J.W., Q.L., L.K., S.Z., and S.O. performed structural studies and analyzed protein binding using biophysical tools. J.L. and S.C. performed the molecular docking. Z.Q.L., J.F., X.L., L.S., S.O., and C.D. interpreted the results. Z.Q.L. wrote the manuscript and all authors provided editorial input.

## Competing interests

The authors declare no competing interests.
