## [Peer Review File · Nature Communications]

Legionella maintains host cell ubiquitin homeostasis by effectors with unique catalytic mechanismsEditorial Note: This manuscript has been previously reviewed at another journal. This document only contains reviewer comments and rebuttal letters for versions considered at *Nature Communications*. Mentions of prior referee reports have been redacted.

REVIEWER COMMENTS

Reviewer #1 (Remarks to the Author):

Fu et al. provide a revised version of their manuscript that describes LnaB as an AMPylator of PR-Ub to create ADPR-Ub, which MavL further processes to produce ADP-ribose and free ubiquitin.

Unfortunately, the authors did not respond to the criticism by conducting additional experiments and strengthening the evidence that their findings are physiologically relevant (e.g., investigating bacterial growth or NF-kappaB signaling).

Thus, the study remains largely based on observations with overexpressed proteins and/or in vitro studies.

Reviewer #2 (Remarks to the Author):

The authors have addressed many of the concerns and the manuscript overall reports an important finding for the Legionella field. Although questions remain, the work provides first steps toward determining the role of MavL and LnaB in the pathogenesis of Legionella. I do not have any additional concerns and recommend publication.

Reviewer #4 (Remarks to the Author):

The authors have significantly improved the manuscript, however they still have not addressed several important concerns. First, there is still a lack of proper data reporting. There are 3 crystal structures but I could only find 2 validation reports. The one for MavL40-404(D315A) ADPR-Ub was missing. Additionally, the authors have still not shown properly calculated 2Fo-Fc and Fo-Fc maps in their figures as requested - either main or supplemental. Finally and most importantly, the authors still refuse to properly cite Voth. Et. Al. They cite the paper but give no context to the findings that paper which they need to do for this manuscript. The biochemical and structural findings directly relate to the study here.

Major concerns:

Past review:

[REDACTED]

Current review:

The authors have not properly cited this paper as requested. All they say in the manuscript is: "Analysis by the HHpred algorithm¹² revealed that the effector MavL(Lpg2526)¹³ appears to harbor a macrodomain involved in recognition of the ADPR moiety¹³." Additionally, they did not point out in the new manuscript where they had supposedly addressed the lack of citation.

It is baffling to this reviewer why the authors will not give proper credit and context to the findings of a previous study that is directly related to their work.

Past review:

[REDACTED]

Current review:

The authors still have not given the stoichiometry or thermodynamic parameters from the ITC data.

Past review:

[REDACTED]

Current review:

The authors appear to have re-refined the structure, but there are still significant issues. First, for the LnaB-actin complex, why is the clash-score so high? Second, and more importantly they have not added the requested maps to the manuscript. When showing a ligand in a crystallographic structure one must provide a 2Fo-Fc map calculated normally and an Fo-Fc map calculated with the ligand removed. These maps need to be contoured to show that the observed density is not biased. The authors say they have shown an "annealing omit-map" contoured at 1.5 sigma. Which map coefficients are used? How was this calculated. I believe this is still a 2Fo-Fc from a simulated annealing calculation?

Also, where is the validation report for the 3rd crystal structure MavL40-404(D315A) ADPR-Ub?

Past review:

[REDACTED]

Current review:

While it is good that the authors have included this citation, I couldn't find any supplemental data showing the results of the full docking search as requested. Energies of the top clusters, groupings of the clusters etc. This is required as previously. and is required for all docking studies.

In the methods the authors say:

"One dominant cluster was identified in the docking calculation. The conformation with the lowest binding energy from the cluster was selected as the predicted complex model."

Why was it the dominant? What criteria was chosen to say this cluster is the dominant? For example, how many times was it predicted compared to other clusters? How many clusters were predicted? What is the energy of this cluster exactly? Being the lowest energy cluster doesn't

guarantee it is correct especially if the energy difference or scores between clusters is low. The results of the top hits and their scores need to be in a supplemental file.

We appreciate the time and efforts by the reviewer in evaluating our work. We have more carefully revise the manuscript to address their comments and suggestions. Our point to point response to their comments is as follows:

Reviewer #1 (Remarks to the Author):

Fu et al. provide a revised version of their manuscript that describes LnaB as an AMPylator of PR-Ub to create ADPR-Ub, which MavL further processes to produce ADP-ribose and free ubiquitin.

Unfortunately, the authors did not respond to the criticism by conducting additional experiments and strengthening the evidence that their findings are physiologically relevant (e.g., investigating bacterial growth or NF-kappaB signaling).

Thus, the study remains largely based on observations with overexpressed proteins and/or in vitro studies.

Response: In response to comments by a reviewer from the last submission, we have removed the results dealing with NFkB activation in the revision. It is perplexing fact widely accepted by the scientific community working on Legionella Dot/Icm effectors that these proteins collectively are important for bacterial intracellular replication, but individually rarely a growth defect was observed when a single gene was deleted. Nevertheless, in light of the instructions by the editor, we will leave the pursuit to their role of MavL and LnaB in intracellular bacterial growth to future study.

:

Reviewer #2 (Remarks to the Author):

The authors have addressed many of the concerns and the manuscript overall reports an important finding for the Legionella field. Although questions remain, the work provides first steps toward determining the role of MavL and LnaB in the pathogenesis of Legionella. I do not have any additional concerns and recommend publication.

Response: We appreciate the positive assessment of our work. Our labs and probably other scientists in the field will continue to pursue the remaining questions in future studies.

Reviewer #4 (Remarks to the Author):

The authors have significantly improved the manuscript, however they still have not addressed several important concerns. First, there is still a lack of proper data reporting. There are 3 crystal structures but I could only find 2 validation reports. The one for MavL40-404(D315A) ADPR-Ub was missing. Additionally, the authors have still not shown properly calculated 2Fo-Fc and Fo-Fc maps in their figures as requested - either main or supplemental. Finally and most importantly, the authors still refuse to properly cite Voth. Et. Al. They cite the paper but give no context to the

findings that paper which they need to do for this manuscript. The biochemical and structural findings directly relate to the study here.

Response: We wholeheartedly appreciate the careful review of our work. These comments have prompted us to more carefully examine our structures and models.

Major concerns:

Past review:

[REDACTED]

Response: We agree with this criticism and have taken the suggestion in our revision by stating that the work by Voth et al provided the hint that MavL is relevant to ubiquitin signaling (together with the fact that our screening was based on the ability of candidate proteins to interact with ubiquitin) (lines 80-87 in the revised manuscript).

Current review:

The authors have not properly cited this paper as requested. All they say in the manuscript is:

“Analysis by the HHpred algorithm¹² revealed that the effector MavL(Lpg2526)¹³ appears to harbor a macrodomain involved in recognition of the ADPR moiety¹³.”

Additionally, they did not point out in the new manuscript where they had supposedly addressed the lack of citation.

It is baffling to this reviewer why the authors will not give proper credit and context to the findings of a previous study that is directly related to their work.

Response: We agree with this criticism and have revised the text in accordance with the reviewer’s comments as we as the fact the work by Voth et al has suggested that MavL is involved in ubiquitin signaling.

The revised text is as follows:

“Among these, MavL(Lpg2526) has been studied in a recent study which reveals that this protein binds ADP-ribose and structural overlay with poly-(ADP-ribose) glycohydrolases (PARGs) identified a pair of aspartate residues potentially involved in catalysis¹². This study also found that MavL interacts with the mammalian ubiquitin-conjugating enzyme UBE2Q1, suggesting a role in ubiquitin signaling¹². In light of these observations and the fact MavL was identified in our screening strategy based on ubiquitin interaction, we further pursued the hypothesis that MavL acts on proteins that have been modified by ubiquitination, ADP-ribosylation or both).

In addition, in our discussion of the residues important for the catalysis of MavL, we have added a sentence to reflect the fact the study by Voth et al has predicted that D323 and D333 are involved in catalysis (line 101-102).

Past review:

Response: We have included the data in the revised manuscript (Figs. S3, S4 and S11).

Current review:

The authors still have not given the stoichiometry or thermodynamic parameters from the ITC data.

Response: Thank you for pointing these issues out. We have included the data in the revised manuscript (Figs. 2C, S3, S4 and S11).

Current review:

The authors appear to have re-refined the structure, but there are still significant issues. First, for the LnaB-actin complex, why is the clash-score so high? Second, and more importantly they have not added the requested maps to the manuscript. When showing a ligand in a crystallographic structure one must provide a 2Fo-Fc map calculated normally and an Fo-Fc map calculated with the ligand removed. These maps need to be contoured to show that the observed density is not biased. The authors say they have shown an “annealing omit-map” contoured at 1.5 sigma. Which map coefficients are used? How was this calculated. I believe this is still a 2Fo-Fc from a simulated annealing calculation?

“First, for the LnaB-actin complex, why is the clash-score so high?”

Response: Thanks for this helpful comment. We have done our best to re-refined the structure of LnaB-actin complex and re-uploaded the validation report.

“Second, and more importantly they have not added the requested maps to the manuscript. When showing a ligand in a crystallographic structure one must provide a 2Fo-Fc map calculated normally and an Fo-Fc map calculated with the ligand removed. These maps need to be contoured to show that the observed density is not biased. The authors say they have shown an “annealing omit-map” contoured at 1.5 sigma. Which map coefficients are used? How was this calculated. I believe this is still a 2Fo-Fc from a simulated annealing calculation? ”

Response: Thank you for the insightful comments. We apologized for not clearly describing the "annealing omit map" in last revision. It is indeed a simulated annealing 2mFo-DFc composite omit map calculated for the MAVL-ADPR and MavL₍₄₀₋₄₀₄₎D315A-Ub-ADPR complexes contoured at 1.5 σ . The omit maps were generated by running the software in the simulated-annealing mode. The annealing method used was cartesian with an annealing temperature of 5,000K.

Per your request, we have provided the new 2mFo-DFc map calculated normally and an mFo-DFc map calculated with the ligand being removed. The 2mFo-DFc map and the mFo-DFc map were calculated using the phenix.FFT_map_coefficients software. The 2mFo-DFc (blue) and mFo-DFc (green) electron-density maps are contoured at the 1.5 σ and 3.0 σ levels, respectively. We have included the maps here below.

Fig 1d (right). The 2mFo-DFc (blue) and mFo-DFc (green) electron-density maps of MavL₍₄₀₋₄₀₄₎D315A-Ub-ADPR were contoured at the 1.5 σ and 3.0 σ levels, respectively.

Supplementary Figure 1e (3). The 2mFo–DFc (blue) and mFo–DFc (green) electron-density maps of MavL-ADPR were contoured at the 1.5 σ and 3.0 σ levels, respectively.

Current review:

While it is good that the authors have included this citation, I couldn't find any supplemental data showing the results of the full docking search as requested. Energies of the top clusters, groupings of the clusters etc. This is required as previously. and is required for all docking studies. In the methods the authors say: "One dominant cluster was identified in the docking calculation. The conformation with the lowest binding energy from the cluster was selected as the predicted complex model." Why was it the dominant? What criteria was chosen to say this cluster is the dominant? For example, how many times was it predicted compared to other clusters? How many clusters were predicted? What is the energy of this cluster exactly? Being the lowest energy cluster doesn't guarantee it is correct especially if the energy difference or scores between clusters is low. The results of the top hits and their scores need to be in a supplemental file.

Response: We'd like to thank the reviewer for pointing these issues out. We have added docking results and more detailed description in the supplementary materials in the reversion. The results of 300 docking runs were grouped into six clusters according to the binding conformations of ATP (Supplementary Fig. 10b). Based on number of conformations and mean binding energy, only one dominant cluster (cluster 1) was identified in the docking calculation. Cluster 1 covered 68.7% of all docking

conformations, and the number of conformations in cluster 1 (206 conformations) was more than five times that in cluster 2 (37 conformations). Furthermore, cluster 1 featured with the lowest mean binding energy (-6.53 ± 0.41 kcal/mol), compared to other clusters (-5.57 ± 0.52 to -3.78 ± 0.48 kcal/mol). Therefore, we selected the conformation with the lowest binding energy in cluster 1 as the putative structural model of the complex (Supplementary Fig. 10b). We have also included the results here.

Supplementary Figure 10b Cluster index of the docking of LnaB with ATP. The numbers of conformations and mean binding energy (in kcal/mol) for each cluster are represented by blue columns and purple dots, respectively. All the docked conformations were clustered using the default parameters implemented in AutoDockTools.

REVIEWER COMMENTS

Reviewer #4 (Remarks to the Author):

The authors are significantly improved the manuscript and have addressed most of my concerns. However, they have still not provided a figure with the correct maps to prove that their ligand is present. Based on the validation report it is 'probably' there, but they need to show an Fo-Fc map after removing the ligand and re-refining. If the ligand is truly there, then there will be a large unmodeled portion of density in the Fo-Fc map in the shape of the ligand. Usually at an rmsd of 2 or higher.

In Figure 1D there only appears to be strong density in the 2Fo-Fc (blue). You cannot see any Fo-Fc density (green). Unless the drawings completely overlap? Typically one presents this data as one panel for the 2Fo-Fc and a second for the Fo-Fc.

Reviewer #4 (Remarks to the Author):

The authors are significantly improved the manuscript and have addressed most of my concerns. However, they have still not provided a figure with the correct maps to prove that their ligand is present. Based on the validation report it is 'probably' there, but they need to show an Fo-Fc map after removing the ligand and re-refining. If the ligand is truly there, then there will be a large unmodeled portion of density in the Fo-Fc map in the shape of the ligand. Usually at an rmsd of 2 or higher.

In Figure 1D there only appears to be strong density in the 2Fo-Fc (blue). You cannot see any Fo-Fc density (green). Unless the drawings completely overlap? Typically one presents this data as one panel for the 2Fo-Fc and a second for the Fo-Fc.

Thanks for the insightful comments. We agree with this criticism and have taken the suggestion in our revision by providing a normal calculated 2Fo-Fc map of **MavL₍₄₀₋₄₀₄₎D315A-Ub-ADPR** (panel 1 below). We have removed the ligand and refined the structure again with Phenix to calculate a new Fo-Fc map (panels 2 and 3 below). The 2Fo-Fc map (blue) and the Fo-Fc map (green) were calculated using the ccp4.Run FFT-Create Map software and contoured at the 1.0σ and 3.0σ levels, respectively. We also show the electron density map (2Fo-Fc and Fo-Fc) and the Fo-Fc map of **MavL₍₄₀₋₄₀₄₎D315A-Ub-ADPR** at an rmsd of 2 and 3 by Coot, respectively (panels 4 and 5). We used panels 1 and 2 in the revised Fig.1D.

The corresponding maps for **MavL₍₄₀₋₄₀₄₎D315A-Ub-ADPR** are also shown below.

1. The 2Fo-Fc map of MavL₍₄₀₋₄₀₄₎D315A-Ub-ADPR was generated with Pymol ($\sigma=1.0$). 2-3. The Fo-Fc maps of MavL₍₄₀₋₄₀₄₎D315A-Ub-ADPR without and with ADPR were generated with Pymol ($\sigma=3.0$). 4. The electron density map of MavL₍₄₀₋₄₀₄₎D315A-Ub-ADPR at an rmsd of 2, generated by Coot. 5. The Fo-Fc map of ADPR at an rmsd of 3, generated by Coot.

The corresponding 2Fo-Fc and Fo-Fc maps of **MavL-ADPR** are calculated using the same method as for MavL₍₄₀₋₄₀₄₎D315A-Ub-ADPR and are shown below. Panels 1 and 2 are used in the revised Fig. S1E.

1. The 2Fo-Fc map of MavL-ADPR was generated with Pymol ($\sigma=1.0$). 2-3. The Fo-Fc maps of MavL-ADPR without and with ADPR were generated with Pymol ($\sigma=3.0$). 4. The electron density map of MavL-ADPR at an rmsd of 2, generated by Coot. 5. The Fo-Fc map of ADPR at an rmsd of 3, generated by Coot.

REVIEWERS' COMMENTS

Reviewer #4 (Remarks to the Author):

The authors have addressed my concerns.